# Nonequilibrium charge-density-wave order beyond the thermal limit

J. Maklar [1✉], Y. W. Windsor [1], C. W. Nicholson [1,7], M. Puppin [1,8], P. Walmsley [2,3], V. Esposito [3,4], M. Porer[4], J. Rittmann[4], D. Leuenberger[5], M. Kubli[6], M. Savoini[6], E. Abreu[6], S. L. Johnson[4,6], P. Beaud [4], G. Ingold[4], U. Staub [4], I. R. Fisher [2,3], R. Ernstorfer [1], M. Wolf[1] & L. Rettig [1✉]

The interaction of many-body systems with intense light pulses may lead to novel emergent phenomena far from equilibrium. Recent discoveries, such as the optical enhancement of the critical temperature in certain superconductors and the photo-stabilization of hidden phases, have turned this field into an important research frontier. Here, we demonstrate nonthermal charge-density-wave (CDW) order at electronic temperatures far greater than the thermo-dynamic transition temperature. Using time- and angle-resolved photoemission spectroscopy and time-resolved X-ray diffraction, we investigate the electronic and structural order parameters of an ultrafast photoinduced CDW-to-metal transition. Tracking the dynamical CDW recovery as a function of electronic temperature reveals a behaviour markedly different from equilibrium, which we attribute to the suppression of lattice fluctuations in the transient nonthermal phonon distribution. A complete description of the system's coherent and incoherent order-parameter dynamics is given by a time-dependent Ginzburg-Landau framework, providing access to the transient potential energy surfaces.

[1] Fritz-Haber-Institut der Max-Planck-Gesellschaft, Berlin, Germany. [2] Geballe Laboratory for Advanced Materials and Department of Applied Physics, Stanford University, Stanford, CA, USA. [3] Stanford Institute for Materials and Energy Sciences, SLAC National Accelerator Laboratory, Menlo Park, CA, USA. [4] Swiss Light Source, Paul Scherrer Institut, Villigen PSI, Switzerland. [5] Department of Physics, University of Zürich, Zürich, Switzerland. [6] Institute for Quantum Electronics, Physics Department, ETH Zürich, Zürich, Switzerland. [7] Present address: Department of Physics and Fribourg Center for Nanomaterials, University of Fribourg, Fribourg, Switzerland. [8] Present address: Laboratory of Ultrafast Spectroscopy, ISIC, Ecole Polytechnique Fédérale de Lausanne (EPFL), Lausanne, Switzerland. ✉email: maklar@fhi-berlin.mpg.de; rettig@fhi-berlin.mpg.de

Complex solids exhibit a multitude of competing and intertwined broken symmetry states originating from a delicate interplay of different degrees of freedom and dimensionality. Among these states, charge-density-waves (CDWs) are a ubiquitous phase characterized by a cooperative periodic modulation of the charge density and of the crystal lattice, mediated by electron-phonon coupling[1–3]. While lattice and charges are intrinsically coupled in equilibrium, ultrafast optical excitation allows to selectively perturb each of these subsystems and to probe the melting of order and its recovery as a real-time process. This approach grants access to the relevant interactions of CDW formation[4–15], to out-of-equilibrium and metastable states[16–19] and elucidates competing orders[20–22].

In close analogy to superconductivity, the formation of a CDW broken symmetry ground state can be described by an effective mean field that serves as an order parameter, which is governed in equilibrium by a static free energy surface. While mean field theory captures the phase transition on a qualitative level, thermal lattice fluctuations reduce the critical temperature $T_c$ of long-range 3D order significantly below the predicted mean field value $T_{MF}$[1,2]. It is of strong interest how our understanding of phase transitions in the adiabatic limit can be adapted to a none-quilibrium, dynamical setting induced by an impulsive excitation[11,23–27]. It remains an open question whether the thermal transition temperature is still a relevant quantity in the description of such an out-of-equilibrium state, and which parameters permit transient control of $T_c$[20,28–32].

Symmetry-broken phases also allow for collective excitations of the order parameter, as observed in a variety of systems, including CDW compounds, superconductors and atoms in optical lattices[33–35]. Two types of modes emerge in the symmetry-broken ground state, related to a variation of the amplitude and the phase of the complex order parameter, i.e., the Higgs amplitude mode (AM) and the Nambu-Goldstone phase mode. In CDW compounds, upon impulsive excitation, the AM manifests as coherent oscillations of the electronic and structural order-parameter amplitudes[4,6,36]. However, recent studies investigating the structural dynamics of various CDW compounds upon strong perturbation hint towards collective modes at increased frequencies far beyond the intrinsic AM[11,26,37].

To address these issues, we investigate the electronic and structural order of optically excited bulk $TbTe_3$, a prototypical CDW compound of the rare-earth tritelluride family[38,39]. Using time- and angle-resolved photoemission spectroscopy (trARPES) in combination with time-resolved X-ray diffraction (trXRD), schematically depicted in Fig. 1a, we extract the amplitude of the electronic and structural order parameters and the electronic temperature as functions of pump-probe delay $t$. This reveals CDW formation at electronic temperatures substantially above the thermal critical temperature. We attribute this transient stabilization to a reduced contribution of lattice fluctuations in the out-of-equilibrium state due to a nonthermal phonon population. Furthermore, with increasing excitation density, the coherent order parameter dynamics indicate a transition from the AM regime to a high-frequency regime, driven by a modification of the underlying potential energy surface. We model the order-parameter dynamics in a time-dependent Ginzburg–Landau framework, which further supports the scenario of a nonthermal stabilization of the CDW order.

## Results

**Electronic and structural CDW signatures.** First, using ARPES, we analyze the Fermi surface (FS) of $TbTe_3$ at $T = 100$ K, well below $T_c = 336$ K, the transition temperature of the unidirectional CDW phase[40]. The electronic properties near $E_F$ are governed by

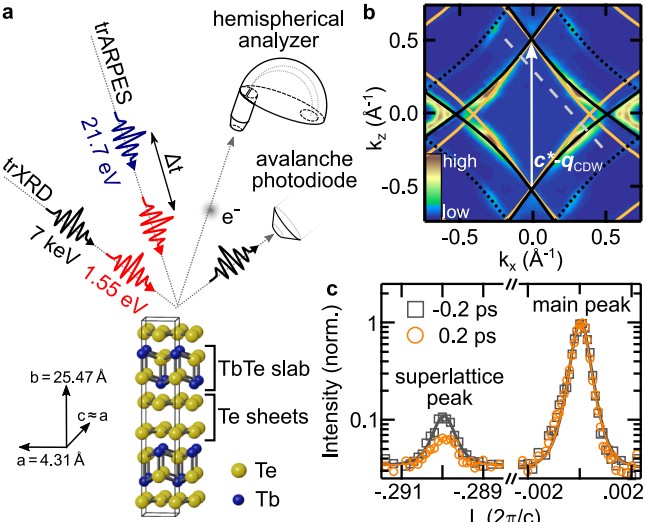

**Fig. 1 Experimental scheme. a** Schematic of the trARPES and grazing-incidence trXRD experiments. $TbTe_3$ is a quasi-2D compound consisting of a stack of Te sheets and TbTe slabs. **b** Symmetrized FS of $TbTe_3$ ($T = 100$ K, $t = 0$ fs). Below $T_c$, the spectral weight within the nested FS regions connected by the CDW wave vector $c^* - q_{CDW}$ vanishes[39]. The black solid and dotted lines correspond to Te $5p_x/5p_z$ bands from tight-binding calculations. FS nesting also leads to the formation of shadow bands (orange lines). The gray dashed line indicates the momentum-direction analyzed in Fig. 2a–c. **c** Representative X-ray Bragg peaks with Voigt fits along the (3 7 L) direction before and after optical excitation (absorbed fluence $F = 1.35$ mJ cm$^{-2}$).

the Te sheets (Fig. 1a), which give rise to the diamond-shaped bands shown in Fig. 1b. Strongly wave-vector dependent electron-phonon coupling[41], in conjunction with a moderately well-nested Fermi surface[42], lead to a unidirectional CDW in which some portions of the Fermi surface are gapped while others remain metallic[39]. To study the effect of the CDW on the lattice, we investigate the intensity of superlattice (SL) Bragg peaks using trXRD. These SL peaks arise from the periodic lattice distortion associated with the CDW, and are displaced by the CDW wave vector $\pm q_{CDW}$ from the main peak positions[40,43]. As Fig. 1c shows, photoexcitation strongly suppresses the SL peak corresponding to a rearrangement of the atomic mean positions towards the trivial metallic phase, while the main lattice peak reflecting the average crystal structure shows only minor changes.

Next, we investigate the electron dynamics associated with the CDW upon photoexcitation. We focus on an energy-momentum cut that contains the electronic signatures of the CDW, namely the energy gap at $E_F$ in the nested regions and the backfolded shadow bands[44], shown in Fig. 2a, b. At temporal pump-probe overlap ($t = 0$ fs), the interacting tight-binding model introduced by Brouet et al.[39] is in excellent agreement with the observed quasiparticle dispersion: In the nested region (left side of Fig. 2a, b), we observe a pronounced hybridization energy gap at $E_F$. In the imperfectly nested region (right side), the Te band exhibits metallic behavior, as the energy gap is located above $E_F$. Furthermore, we observe faint shadow bands in the vicinity of the energy gaps (boxes 2 and 3 in Fig. 2b). Within 120 fs, the system undergoes a photo-induced CDW-to-metal transition[6], as apparent from the transient suppression of the energy gap and the shadow bands, see Fig. 2c–e.

**CDW order-parameter dynamics.** The CDW-to-metal transition can be described by an order parameter $\psi$, with $|\psi| = 0$ in the metallic and $0 < |\psi| \leq 1$ in the CDW phase. Due to the coupling

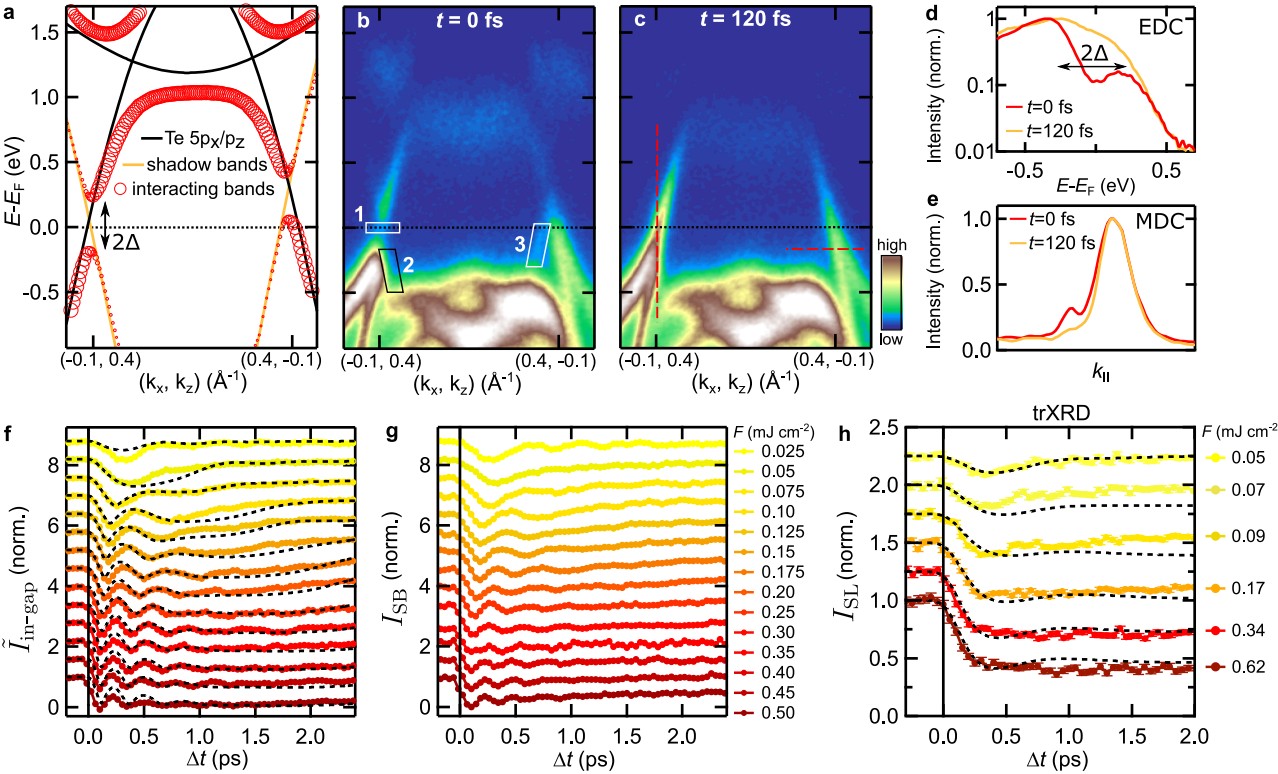

**Fig. 2 CDW band structure dynamics. a** Tight-binding bands along the momentum-direction indicated by the dashed gray line in Fig. 1b. The black and orange curves correspond to the non-interacting Te main and shadow bands, respectively. The red circles mark the hybridized bands with interaction potential Δ. The circle size illustrates the spectral weight. **b, c** trARPES measurements ($F = 0.45$ mJ cm$^{-2}$) along the momentum direction shown in **a**. At $t = 0$ fs, the energy gap at $E_F$ (box 1) and shadow bands (boxes 2, 3) indicate the CDW order. After 120 fs, the CDW vanishes, and the energy gap and shadow band intensity are strongly suppressed. **d-e** Energy and momentum distribution curves along the dashed vertical and horizontal lines in **c**, respectively. **f** Inverted in-gap intensity $\tilde{I}_{\text{in-gap}} = 1 - I_{\text{in-gap}}$ with in-gap intensity $I_{\text{in-gap}}$ (box 1 in **b**, normalized by the respective pre-excitation values) as function of pump-probe delay for various fluences (displaced vertically). Normalized time-dependent Ginzburg–Landau simulations are shown in black. For details of the model, see main text and Supplementary Note 1. **g** Normalized shadow band intensity extracted from box 2. The shadow band intensity obtained from box 3 is shown in Supplementary Fig. 1. **h** Time evolution of the $(2\,10\,1 + q_{\text{CDW}})$ SL peak intensity for various fluences (displaced vertically) with layered Ginzburg–Landau simulations, see Supplementary Note 3. The curves are normalized by their respective pre-excitation values. The error bars correspond to one standard deviation from photon counting statistics.

between charges and lattice, the transition can be characterized by an electronic ($\psi_e$) or a structural ($\psi_s$) order parameter. We utilize trARPES to access the amplitude of the electronic order parameter $|\psi_e|$. Most directly, $|\psi_e|$ can be extracted by tracking the energy gap $2\Delta$ at $E_F$[13,45]. However, this method faces practical limitations due to the vanishing occupation of bands above $E_F$ after a few 100 fs and due to the limited experimental energy resolution. Therefore, we choose two alternative metrics to quantify the CDW order: We introduce the inverted in-gap intensity $\tilde{I}_{\text{in-gap}} = 1 - I_{\text{in-gap}}$ with normalized in-gap intensity $I_{\text{in-gap}}$, extracted from box 1 in Fig. 2b. We find that this metric – for the chosen region of interest and our experimental resolution – follows a BCS-like temperature dependence in equilibrium, as confirmed by static measurements (black markers in Fig. 3b), and thus is considered equivalent to $|\psi_e|$. Further, as the inverted in-gap intensity is derived from a region where the gap is centered around $E_F$, it is unaffected by thermal changes to the distribution function. As a second metric, we extract the shadow band intensity $I_{\text{SB}} \propto |\psi_e|$[30,44] from box 2 in Fig. 2b.

Using these equivalent metrics, we investigate the photo-induced CDW suppression and recovery over a wide range of fluences, as shown in Fig. 2f–g. For a low absorbed fluence of 0.025 mJ cm$^{-2}$ below the CDW melting threshold, we observe a weak modulation of the CDW gap and SB intensity corresponding to the AM of the CDW at $\omega_{\text{AM}}/2\pi = 2.2$ THz (see

Supplementary Fig. 2). At the CDW melting threshold $\approx 0.05$ mJ cm$^{-2}$, the AM softens and becomes overdamped, while the CDW melting time $t_{\text{melt}}$ slows down, and the energy gap and SB intensity vanish almost completely. Upon crossing the melting threshold, we observe a fast initial quench of the CDW within $t_{\text{melt}} \approx 100$ fs (see Supplementary Fig. 8), followed by few damped coherent oscillations that exhibit a pronounced frequency reduction with pump-probe delay (down-chirp). Interestingly, the initial frequency of the collective excitation increases with fluence, doubling at the highest accessible fluences. Concurrently, the time required to restore the ground state after perturbation steadily increases with fluence, leading to a persistent suppression of the CDW for a few ps at the highest excitation densities we used.

To gain a complementary view of the photo-induced phase transition, we use trXRD to extract the structural order parameter from the normalized SL peak intensity upon optical excitation[11,23,37], which, in first approximation, is given by $I_{\text{SL}}(t) \propto |\psi_s(t)|^2$. As Fig. 2h shows, the SL response qualitatively resembles the dynamical quench and recovery of the extracted electronic order parameter. In the low-fluence regime, a weak initial suppression is followed by a quick recovery of the SL structure, on top of which a faint modulation can be identified (see Supplementary Fig. 3). In the high-fluence regime, the SL peak intensity is strongly quenched, and, with increasing fluence,

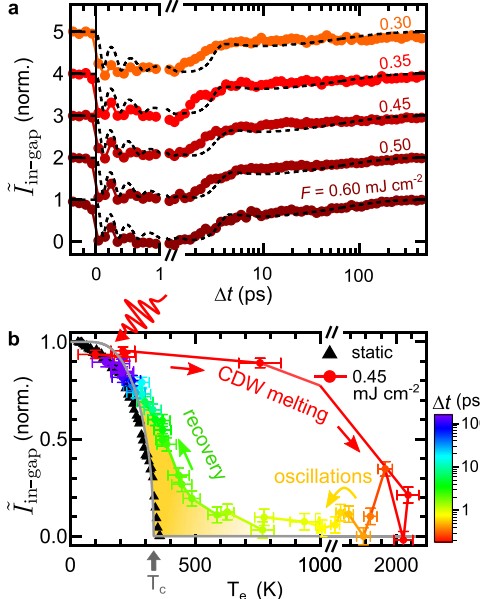

**Fig. 3 CDW recovery dynamics. a** Time evolution of the inverted in-gap intensity in the high-fluence regime (displaced vertically). Normalized time-dependent Ginzburg–Landau simulations are shown in black. **b** Inverted in-gap intensity versus extracted electronic temperatures. One standard deviation of the $T_e$ fit (horizontal error bars) and one standard deviation derived from electron counting statistics (vertical error bars) are given as uncertainty. $\tilde{I}_{\text{in-gap}}$ extracted from a static temperature series (black markers, $T$ values from heater setpoints, curve normalized to the lowest accessible $T$ value) is in general agreement with the BCS-type $T$-dependence of the order parameter (gray curve). The dynamic trace shows the full cycle of laser-heating and CDW melting, coherent oscillations and CDW recovery (delay encoded in the color code). The yellow shaded area marks the region of dynamical CDW formation at electronic temperatures above $T_c$. The pre-excitation value of the dynamic trace ($T = 100$ K) is normalized to the corresponding value of the static $T$-dependence.

the time required to recover diverges. In contrast to the electronic response, we do not observe clear coherent oscillations of the SL peak intensity upon strong excitation. This originates most likely from the lower temporal resolution of the trXRD setup and the contribution of sub-surface crystal layers with varying, lower excitation densities (see Supplementary Note 3). Recent trXRD experiments with improved temporal resolution have revealed fluence-dependent collective excitations of the SL peak intensity in a closely related tritelluride[37] – in agreement with our observations for $\psi_e$. Furthermore, while the SL intensity $I_{\text{SL}}$ drops linearly with excitation density shortly after excitation, this behavior plateaus after crossing a fluence of $\approx 0.1$ mJ cm$^{-2}$. This results in a residual SL intensity of 35% even after strong excitation of up to 1.35 mJ cm$^{-2}$. We assign this persisting SL background to a contribution of unexcited sample volumes due to surface steps caused by crystal cleaving[11]. Nonetheless, the trXRD data clearly shows that not only the electronic, but also the lattice superstructure is melted upon strong photoexcitation. The qualitative agreement of the electronic and structural response demonstrates a strong coupling between electronic and lattice degrees of freedom on ultrafast timescales, and suggests an equivalent treatment of $|\psi_s|$ and $|\psi_e|$ within the experimental time resolution.

Diffraction also probes the long-range coherence of the SL phase. While phase coherence plays a secondary role in the low-fluence regime, it becomes increasingly important during the CDW recovery after strong perturbation due to the creation of

topological defects. These dislocation-type defects broaden the SL peaks, locally decrease the amplitude of the periodic lattice modulation, and can persist long after the CDW amplitude has recovered[46–48]. Therefore, rather than trXRD, we employ trARPES to access the amplitude of the order parameter throughout the full recovery to equilibrium. As shown in Fig. 3a, in the high-fluence regime, the majority of the CDW order is restored after $\approx 5$ ps, followed by a complete recovery on a 100 ps timescale.

**Transient electronic temperature**. Time-resolved ARPES allows to extract the transient electronic temperatures from Fermi–Dirac fits to the energy distribution of metallic regions of the FS (see Supplementary Note 2), and thereby to compare the none-quilibrium CDW melting and recovery to the mean field behavior upon thermal heating. Remarkably, in the dynamic case, the electronic order parameter does not follow the mean field dependence governed by $T_c$. In the low-fluence regime below the CDW melting threshold, electronic temperatures reach up to 500 K, far above $T_c = 336$ K (see Supplementary Fig. 7). Yet, photo-excitation causes only a minor initial suppression of the energy gap and of the periodic lattice distortion, and initiates a collective AM oscillation – a hallmark of the CDW state.

In the high-fluence regime, the CDW is fully suppressed ($\tilde{I}_{\text{in-gap}} = I_{\text{SB}} = 0$) as initial electronic temperatures exceed 2000 K. However, recovery of the CDW order already sets in when the electronic system is still at elevated temperatures $T_e \gg T_c$. To illustrate this dynamic behavior, Fig. 3b presents the inverted in-gap intensity of the melting and the recovery cycle as a function of extracted electronic temperatures. In the out-of-equilibrium setting, CDW order reappears below $T_e \approx 600$ K (yellow shaded area), indicating an increased effective critical temperature $T_c^*$. At delay times of several ps, corresponding to electronic temperatures of $T_e \leq T_c$, the dynamic behavior converges to the equilibrium $T$-dependence. This trend of nonthermal CDW recovery is consistent over a wide range of fluences (see Supplementary Fig. 4).

**Time-dependent Ginzburg–Landau theory**. Near the transition temperature, the order parameter can be approximated by the Landau theory of second-order phase transitions[2]. Thus, to simulate the dynamics of the order parameter in TbTe$_3$, we make the following ansatz for the effective potential energy surface (in dimensionless units) based on time-dependent Ginzburg–Landau (tdGL) theory[11,27,37,49,50]:

$$V(\psi, t) = -\frac{1}{2}\left(1 - \eta(t)\right)\psi^2 + \frac{1}{4}\psi^4 . \tag{1}$$

Upon perturbation, the dynamics of the order parameter are determined by the equation of motion derived from Eq. (1) (see Supplementary Note 1). The transient modification of the potential, resulting from the laser excitation and subsequent relaxation, is modeled by the ratio of the electronic temperature and the critical temperature $\eta(t) = T_e/T_c$. Motivated by the increased transient ordering temperature discussed above, we replace the static $T_c$ by a phenomenological time-dependent critical temperature

$$T_c^*(t) = T_c(1 + H(t) \cdot s \cdot \exp(-t/\tau_{\text{ph-ph}})) , \tag{2}$$

with Heaviside step function $H$. It captures the enhanced critical temperature in the nonthermal regime, given by the temperature scaling $s$, and converges to $T_c$ at late times. This leaves us with only two global fit parameters for the simulations: damping $\gamma$ and scaling $s$ in the nonthermal regime (see Supplementary Note 1 for details of the model). For the timescale connecting both regimes,

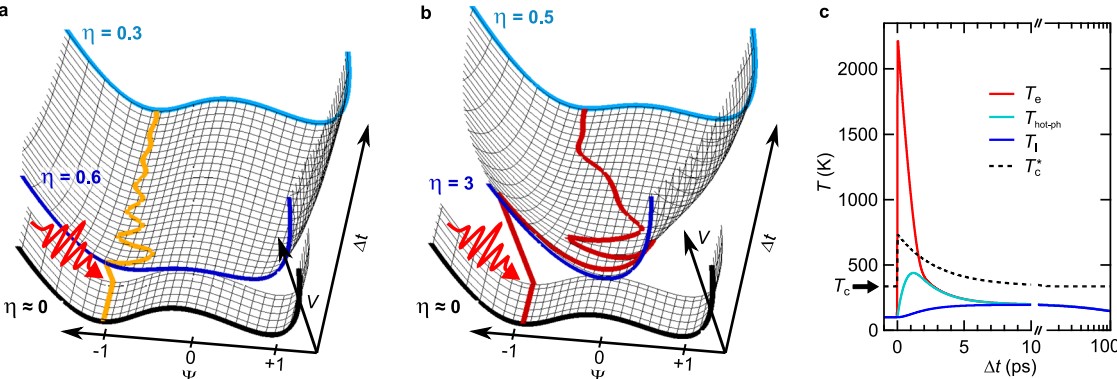

**Fig. 4 Simulated order-parameter dynamics and 3TM.** Transient potential energy surface and order-parameter pathway upon **a** weak and **b** strong optical excitation. The potential shapes before excitation (black curve), at 0 ps (dark blue) and 3.5 ps (light blue) are highlighted. **a** In the AM regime, the double-well potential is weakly modified, while in **b** the overshoot regime, the CDW melting threshold is reached, resulting in a single-well shaped potential, followed by a relaxation to the double-well ground state. **c** 3TM of electronic, hot phonon and lattice temperatures $T_e$, $T_{hot-ph}$ and $T_l$ in the regime of strong perturbation ($F = 0.35$ mJ cm$^{-2}$). In the 3TM, the optical excitation of the electronic system is followed by an energy transfer to certain strongly coupled optical phonons, widely observed in materials with selective electron-phonon coupling[27,51,52,61–63]. Subsequently, this hot phonon subset equilibrates with the remaining lattice phonon bath on a ps timescale ($\tau_{ph-ph}$). To account for the recovery of the base temperature via heat diffusion on a 100 ps timescale, the lattice is coupled to an external heat sink. The black dashed line indicates the rescaled critical temperature $T_c^*$. In the 3TM simulations, material properties of the related compound LaTe$_3$[27] were used.

we find a good description of the data by choosing the lattice thermalization time $\tau_{ph-ph} = 2.2$ ps reported for the closely related compound LaTe$_3$[27]. Energy redistribution processes within the electron and lattice systems are often modeled by a three temperature model (3TM)[51,52], as presented in Fig. 4c. Here, $\tau_{ph-ph}$ corresponds to the timescale of energy transfer between strongly coupled optical phonon modes ($T_{hot-ph}$) with the remaining cold lattice modes ($T_l$). The choice of the parameter $\tau_{ph-ph}$ is further motivated in the following discussion. In this description, CDW order emerges when the electronic temperature $T_e$ falls below the introduced dynamic effective $T_c^*$ (black dashed curve in Fig. 4c). During the thermalization process, the estimated lattice temperatures $T_l$ stay below the thermal critical temperature for all applied fluences.

Given the complexity of the system, this model with its minimal amount of free parameters is in remarkable agreement with the electronic order parameter extracted directly from the trARPES data throughout the CDW melting and full recovery over a large fluence range, as shown in Figs. 2f and 3a. It captures (i) the AM in the low-fluence regime, (ii) the CDW melting time after arrival of the pump, (iii) the coherent oscillations and the down-chirp in the high-fluence regime, and (iv) the full CDW recovery to equilibrium. The fit yields a nonthermal critical temperature of $T_c^*(t = 0$ fs$) \approx 745$ K, i.e., more than double of the equilibrium $T_c$. Remarkably, this value is similar to the electronic temperature where the onset of CDW recovery is observed in Fig. 3b. To illustrate the necessity of a transiently enhanced $T_c^*$ to describe the data, we perform tdGL simulations keeping the critical temperature fixed at the equilibrium value, which, however, leads to a severe deviation from the experimental oscillations and CDW recovery, see Supplementary Fig. 5. Next, we illustrate the characteristic regimes of the tdGL simulations based on the extracted transient potential energy surfaces $V(\psi, t)$ in Fig. 4.

AM regime: Before excitation, the system is in the CDW ground state ($\eta \approx 0$), corresponding to an underlying double-well potential with minima at $|\psi| \approx 1$. Upon weak excitation (Fig. 4a), the potential surface is barely altered and maintains its double-well shape. This launches a damped oscillation of the order parameter around the marginally shifted potential minimum at frequency $\omega_{AM}$, i.e., the AM.

Overshoot regime: Upon strong excitation (Fig. 4b), the underlying potential transforms to a single-well shape, corresponding to the metallic phase. The order parameter overshoots to the opposite side of the potential, and oscillates around the new potential minimum at $|\psi| = 0$ at frequency $\omega \gg \omega_{AM}$. Relaxation of the system leads to a transient flattening of the potential, resulting in the observed frequency down-chirp. At $\eta < 1$, the CDW order finally recovers, and the order parameter relaxes into one of the minima of the emerging double-well potential.

A minor deviation of the fit from the data occurs at the dynamical slowing-down of the CDW melting in the vicinity of the melting threshold, as observed in the curve at fluence 0.05 mJ cm$^{-2}$ in Fig. 2f. For an initial perturbation in the range $\eta_{init} \approx 0.5\ldots1$, the system gains just enough energy to reach the local maximum of the double-well potential at $|\psi| = 0$. Close to this metastable point, the potential is rather flat, leading to a critical slowing-down of the order-parameter dynamics[53], discussed in detail in Supplementary Note 4. A similar critical behavior is expected during the recovery of the CDW order. In the overshoot regime, after dampening of the initial oscillations, the order parameter can get trapped at the metastable local maximum despite an incipient recovery of the double-well ground state. However, in real systems, several microscopic processes, such as local modification of $T_c$ by crystal defects[54,55], CDW nucleation and creation of topological defects[47] and coupling of the collective excitation to other phonons[36], will screen against a pronounced critical slowing-down. However, such effects go beyond our current model.

To reproduce the main observations of the extracted structural order parameter, we extend this model to a layered description (see Supplementary Note 3), as shown in Fig. 2h. However, the absence of clear coherent modulations in the time evolution of the SL peak intensity and the additional contribution of the SL phase coherence prohibit a reliable fit of $I_{SL}(t)$. Nonetheless, we conclude that this model captures all key features of the structural and electronic order parameters within a unified framework.

## Discussion

We unambiguously demonstrate a transient CDW behavior distinct from equilibrium, as evidenced by the CDW AM

modulations after weak excitation despite electronic temperatures exceeding thermal $T_c$, and from the CDW recovery at elevated electronic temperatures after strong excitation. The qualitative correspondence of charge and structural features of the CDW excludes a scenario in which only the electronic superstructure is destroyed while the lattice distortion remains intact, which could facilitate such a nonthermal behavior. So what causes this enhanced transient stability of CDW order far beyond the equilibrium $T_c$? In equilibrium, lattice fluctuations induced by thermally populated phonons, accompanied by fluctuations of the charge density, reduce $T_c$ significantly below the mean-field value $T_{MF}$. Especially in low-dimensional systems, these fluctuation effects become increasingly important, such that long-range order and phase transitions cannot occur at finite temperatures in strictly 1D systems[1,2]. However, in real materials, coupling between neighboring chains stabilizes the CDW order, resulting in short-range correlations at high-temperatures and long-range 3D order below $T_c$[2,3].

Ultrafast optical perturbation breaks the thermal equilibrium between charges and lattice. Initially, electrons and certain optical phonons are strongly excited, while the overall vibrational population of the lattice – determined by acoustic modes that account for the majority of the lattice heat capacity – is still close to its pre-excitation value corresponding to an effective lattice temperature significantly below $T_c$. In this out-of-equilibrium regime, the average displacement of the ionic cores around their mean positions (mean-squared displacement) is small, as the nonthermal phonon population is dominated by high-frequency, low-amplitude optical phonons[56]. Thus, initially after excitation, lattice fluctuations are strongly suppressed and counteract a mean-field long-range ordering only weakly, which facilitates CDW formation even at electronic temperatures far beyond $T_c$, illustrated in Fig. 5. In this nonthermal regime, $T_c$ is replaced by the effective electronic critical temperature $T_c^*$, which is

renormalized towards the mean field value depending on the transient lattice temperature and concomitant fluctuations. Over the course of several ps, depending on the lattice thermalization time $\tau_{ph-ph}$, energy is transferred from the strongly coupled optical hot phonons to the remaining phonon modes. This defines the crossover from the nonthermal to the quasi-thermal regime, at which electrons and lattice locally reach thermal equilibrium. As the lattice temperature rises, acoustic (high-amplitude) fluctuations and CDW phase fluctuations increase, which impedes long-range 3D CDW order, and $T_c^*$ consequently converges towards the equilibrium $T_c$. The increasing occupation of lattice vibrations also increases the lattice entropy, and thus modifies the underlying free energy surface. In this picture, the changing lattice entropy plays the analogous part to the time-dependent critical temperature introduced within our tdGL expansion.

The agreement of the Ginzburg–Landau simulations with the extracted order parameters further underlines this scenario. The initial oscillation frequency of the electronic order parameter, the down-chirp as well as the recovery are reproduced by simulations with an enhanced $T_c^*$, that converges towards the equilibrium $T_c$ on the lattice thermalization time. In addition, since the initial lattice temperature is close to its equilibrium value also after strong excitation to the overshoot regime, the contribution of thermal fluctuations is expected to be rather independent of fluence. This is in agreement with our model, which captures the experimental data over a wide fluence range with a fluence-independent description of $T_c^*$. Our simulations yield a transient critical temperature of $\approx 750$ K at early times, which is still considerably below the mean field transition temperature $T_{MF} \approx 1600$ K estimated from the electronic energy gap in the nested regions via the well-known BCS expression[2]. However, because of the imperfect nesting of large segments of the FS, a significant reduction of $T_{MF}$ is expected[40,57], and remaining fluctuations at the initial lattice temperature of $T_l \approx 100$ K are further expected to lead to a lower $T_c^*$.

The CDW order above $T_c$ may be further stabilized by transiently enhanced FS nesting. A previous trARPES study has demonstrated an improved nesting condition in rare-earth tritellurides upon optical excitation[13], caused by a transient modification of the FS. Consequently, the CDW-gapped area at $E_F$ expands and the critical temperature transiently increases. However, the photo-induced enhanced nesting significantly increases with excitation density[13], which would result in a strongly fluence-dependent nonthermal critical temperature. As we find a good description of the data by $T_c^*$ independent of fluence, we assign a suppression of lattice fluctuations in the out-of-equilibrium state as the dominant effect stabilizing the transient CDW. Several studies suggest similar nonthermal behavior in other CDW materials. The commensurate CDW phase of 1T-$TaS_2$ exhibits an exceptionally robust AM after strong perturbation, with initial electronic temperatures exceeding 1300 K[5]. In elemental Chromium, trXRD measurements of the SL peak indicate a persisting CDW state above the thermal transition temperature[29].

In summary, we experimentally track the structural and electronic order parameters of a photo-induced CDW-to-metal transition in the rare-earth tritelluride $TbTe_3$, and reveal a close correspondence of the charge and lattice components of the CDW phase throughout the melting and initial recovery of order. By extracting the time-dependent electronic temperature, we demonstrated nonthermal CDW formation at electronic temperatures significantly above the thermodynamic transition temperature $T_c$. We attribute the dominating role of this behavior to reduced lattice fluctuations compared to a scenario in which charges and lattice are in equilibrium above $T_c$. Since lattice

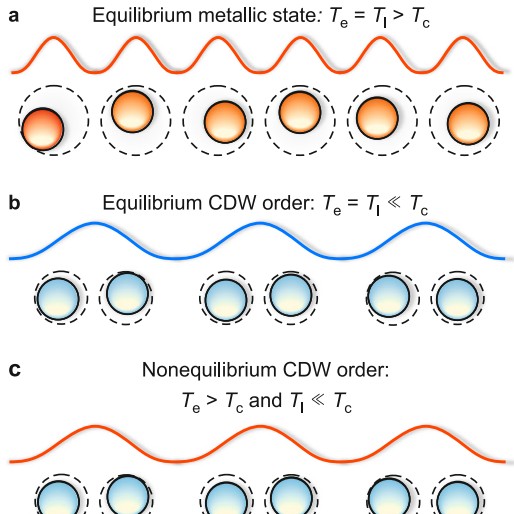

**Fig. 5 Illustration of nonthermal CDW order. a** In equilibrium at elevated temperatures, the system is in a trivial metallic phase. The charge density (wavy line) and the mean positions of the ionic cores (circles) are spaced evenly, as strong thermal lattice fluctuations prevent long-range CDW order. **b** In equilibrium at low temperatures, the system features an ordered charge- and lattice superstructure. **c** Photoexcitation of the CDW ground state ($T_{pre-exc.} \ll T_c$) generates a hot electron distribution, while the lattice initially remains cold. In this out-of-equilibrium state, thermal lattice fluctuations are weak and barely hinder long-range CDW ordering. Hence, the charge and lattice superstructure is stabilized at electronic temperatures beyond $T_c$.

fluctuations play a universal role in the CDW formation, the observed nonthermal stabilization mechanism should also apply to other material families. Moreover, we observed excitation-dependent collective dynamics of the charge order, closely connected to a coherent modulation of the periodic lattice distortion. We applied a tdGL framework to model the order-parameter dynamics and to describe the underlying transient potential energy surface, which governs the collective behavior. Despite its simplicity of using a single degree of freedom, this phenomenological model reproduces all key observations. This suggests that mode-coupling[36] and inhomogeneities (defects) play only a secondary role in the dynamical melting and recovery of the CDW amplitude.

As any memory device relies on nonequilibrium properties, our results have strong implications for applications involving charge-ordering phenomena. A key parameter defining the persistence of the nonthermal stabilization is phonon-phonon coupling, as it dictates the lattice thermalization and thus the timescale on which the fluctuation background rises. Therefore, minimizing phonon-phonon coupling may be critical in the design of switchable CDW devices operating in nonequilibrium conditions[58].

## Methods

**trARPES**. Single crystals of $TbTe_3$ samples were grown by slow cooling of a binary melt[38]. All experiments were carried out at $T = 100$ K. The ARPES measurements were performed in ultra-high vacuum $< 1 \times 10^{-10}$ mbar (samples cleaved in-situ), using a laser-based higher-harmonic-generation trARPES setup[59] ($h\nu_{probe} = 21.7$ eV, $h\nu_{pump} = 1.55$ eV, 500 kHz repetition rate, $\Delta E \approx 175$ meV, $\Delta t \approx 35$ fs) and a SPECS Phoibos 150 hemispherical analyzer. The pump and probe spot sizes (FWHM) are $\approx 230 \times 200$ $\mu m^2$ and $\approx 70 \times 60$ $\mu m^2$. All discussed fluence values refer to the absorbed fluence $F_{abs}$. To estimate $F_{abs}$, the complex refractive index was determined via optical reflectivity measurements at $\lambda = 800$ nm to $n = 0.9$ and $k = 2.6$.

**trXRD**. The trXRD measurements were carried out at the FEMTO hard X-ray slicing source (X05LA) at the Swiss Light Source, Paul Scherrer Institut, Villigen, Switzerland[60]. The utilized laser-sliced X-ray pulses ($h\nu_{X-ray} = 7$ keV, $\Delta t \approx 120$ fs) feature the high stability of conventional synchrotron radiation and do not exhibit any relevant jitter in position, angle or wavelength. The diffracted X-ray intensity was recorded with an avalanche photodiode in an asymmetric diffraction geometry. A synchronized optical pump laser ($10°$ angle of incidence, $h\nu_{pump} = 1.55$ eV, $\Delta t \approx 110$ fs) was used to excite the system. The pump and probe spot sizes (FWHM) were $\approx 600 \times 600$ $\mu m^2$ and $\approx 250 \times 5$ $\mu m^2$. The X-ray extinction length was matched to the pump penetration depth of 25 nm by using a grazing angle of incidence of $0.5°$.

## Data availability

The data that support the findings of this study are publicly available in Zenodo[64] with the identifier https://doi.org/10.5281/zenodo.4106272.

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

## Acknowledgements

We thank E.M. Bothschafter for support during the trXRD experiments. This work was funded by the Max Planck Society, the European Research Council (ERC) under the European Union's Horizon 2020 research and innovation program (Grant no. ERC-2015-CoG-682843), the German Research Foundation (DFG) within the Emmy Noether program (Grant no. RE 3977/1), and the DFG research unit FOR 1700. Crystal growth and characterization at Stanford University (P.W. and I.R.F.) was supported by the Department of Energy, Office of Basic Energy Sciences under Contract no. DE-AC02-76SF00515. Part of this work was supported by the NCCR Molecular Ultrafast Science and Technology (Grant no. 51NF40-183615), a research instrument of the Swiss National Science Foundation (SNSF). E.A. acknowledges support from the ETH Zurich Post-doctoral Fellowship Program and from the Marie Curie Actions for People COFUND Program.

## Author contributions

Y.W.W., L.R., M.Pu., C.W.N. and J.M. carried out the trARPES experiments; L.R., V.E., M.Po., J.R., D.L., M.K., M.S., E.A., S.L.J., P.B., G.I. and U.S. carried out the trXRD experiments; P.W. and I.R.F. provided the samples; J.M. analyzed the data with support from L.R.; J.M. wrote the manuscript with support from L.R., R.E. and M.W.; all authors commented on the paper.

## Funding

## Competing interests

The authors declare no competing interests.
