## [Peer Review File · Nature Communications]

REVIEWER COMMENTS

Reviewer #1 (Remarks to the Author):

The authors present a very interesting study on the non-equilibrium stabilization of a symmetry broken phase due to the transient suppression of low-frequency lattice fluctuations. I am particularly captivated by the extremely convincing application of the Ginzburg-Landau theory for the quantitative description of the experimentally observed non-equilibrium dynamics. The concept is novel and is obviously capable of providing a comprehensive and consistent interpretation of nonequilibrium processes associated with the destruction and formation of broken symmetry states of matter. In my opinion, it has the potential to become in the future a standard tool for the analysis and interpretation of experimental data in this research field.

The presented experimental data is of very high quality and the analysis of the data has been performed with care. The interpretation of the result is sound and strongly supported by the simulations (see my comments above). Overall I strongly recommend the manuscript for publication in Nature Communications. However, before a final decision the authors should address the following points:

The authors introduce the inverted in-gap intensity at EF as a metric for the electronic order parameter because of practical limitations in determining the energy gap. I am wondering how changes in the thermal population of states near EF affect this metric as the electron temperature increases. I suspect that these effects are small at sufficiently low temperatures, i.e. under thermal equilibrium conditions. However, in the experiments transient electron temperatures of up to 2300 K are observed.

The authors mention a CDW melting threshold of 0.1 mJ/cm². It is not obvious to me, how this value was determined from the experimental data. This point should be clarified in a revised version.

The authors furthermore write that "at the melting threshold...the CDW melting time t_{melt} slows down". Actually, Fig S7 implies that the slowing down of t_{melt} (as determined from the trARPES data) stops already way below the CDW melting threshold (at approx. 0.05 mJ/cm²) and that this conclusion is based only on a single data point (which makes this conclusion somehow questionable if only based on the trARPES data). Notably, the trXRD data (Fig. 2f) confirms a slowing down of t_{melt} up to a fluence of approx. 0.1 mJ/cm². So there seems to be some mismatch between the trARPES data and the trXRD data. I suspect that this difference arises from the difference in the probing depth of the two techniques (see second paragraph on page 8). Overall, I feel that this part shows in the present form some inconsistencies. I therefore suggest to revise this paragraph correspondingly.

The agreement between experimental data and simulations based on the Landau-Ginzburg model is indeed excellent, particularly as only two fitting parameters are used. On notices some differences in the 1 to 10 ps regime, where the experimental data show a much smoother recovery than the simulations (here, the logarithmic presentation may be a little bit misleading). Could the authors comment on the origin of the more step-like recovery in the simulation. Is there any obvious reason, why the match between experiment and simulations improves as the fluence increases?

Based on the introduced model, is it possible to somehow extrapolate to an initial lattice temperature $T_I = 0\text{K}$? This could provide valuable information on how much lattice fluctuations contribute to a reduction of T_C in comparison to T_{MF} .

Reviewer #2 (Remarks to the Author):

The authors combine time-resolved ARPES with time-resolved x-ray diffraction for studying charge density wave dynamics in TbTe₃. They demonstrate nonthermal CDW order, existing far above the

critical temperature measured in equilibrium. With tr-ARPES, they show how the CDW order recovers at temperatures higher than the equilibrium critical temperature. For explaining the experimental observations, the authors propose a phenomenological description where the transition temperature, out-of-equilibrium, changes transiently. While the tr-ARPES data is of superb quality, the modeling unfortunately only poorly reproduces the experimental observations. The proposed time-dependent, non-equilibrium critical temperature is an interesting idea. Yet, the choice of the lattice temperature as the origin is insufficiently explained, and the underpinning mechanisms remain unclear. Of course, the critical temperature is known to change with pressure, doping, or disorder – all phenomena that rival the newly proposed mechanism. The recovery timescales predicted by the model disagree with the data by a factor of two in Figure 3a. This disagreement hints at an underlying flaw in the model.

This paper's central point is that the CDW recovers at electronic temperatures above the critical temperature when away from equilibrium. I believe, to convince the reader that this is indeed the point, Figure 3 deserves a revision.

- a. In Fig. 4b, at high temperatures, the inverted in-gap intensity is as high as 0.3. Is this the uncertainty of the measurement (similar scatter is visible in Fig S2)? If yes, then all values above 400 K are within uncertainty consistent with the equilibrium curve.
- b. It is hard to see what colors represent what time delays in Fig. 3b.
- c. In Figure 3b, the BCS curve seems to be distinctly different from the data for temperatures below 200 K. Do the authors have an explanation why their new method of determining the bandgap deviates from BCS?
- d. In Fig. 3a, the dashed line reproduces the experimental data well for large fluences; however, the recovery time appears to differ by a factor of two for lower fluence.

It is somewhat unclear what the tr-X-ray data contributes to the discovery. It is mentioned but not implemented into the discussion beyond that it is qualitatively comparable to tr-ARPES.

How did the authors calculate the absorbed fluence for both experiments?

The pulse durations in both measurements appear to be significantly different. 35 fs in tr-ARPES and 110 fs in time-resolved XRD. Since the authors observe dynamics on a 100 fs time scale, how does the difference modify the results?

The Landau model fit in Figure 2f is well for intermediate fluences; however, it significantly deviates from the data for low and high fluences. Also, in Figure 2h, the model fits substantially differ from the data, well outside the depicted uncertainties. The manuscript deserves a discussion on the discrepancy between the model and the data.

In Fig. 2h, the authors display photon counting statistics. What about FEL jitter in position, angle, and wavelength? Are these uncertainties negligible?

Minor comments

In Fig. 2 b, the indication for box 3 is missing.

In Figure 5 it is difficult to see the point. How would the picture look for an equilibrium CDW order state and how would that picture differ from the non-equilibrium CDW order state.

Reviewer #3 (Remarks to the Author):

An interesting time-resolved ARPES study of transient melting and recovery of charge density wave (CDW) order in TbTe₃ is presented. The authors follow the characteristic CDW-related energy gap and shadow bands as a function of time after optical excitation and repeat the experiment for a broad variety of pump fluences. From these data they extract electronic quasi-temperatures as a function of delay time and fluence and find that, during the electronic cool

down, the CDW gap opens already at electron temperatures distinctly above the equilibrium transition temperature T_c . They explain this scenario using the Ginzburg-Landau theory and a three-temperature model and suggest that after optical excitation the crystal lattice remains much cooler than T_c and, thus, phonon fluctuations do not destabilize the CDW order. Under these conditions, even rather hot electrons may form a CDW. The manuscript also features trXRD data taken under comparable conditions.

Overall, this is obviously a very carefully executed study which uses some of the most sophisticated, expensive and exclusive experimental techniques available at this time. The data have been treated with care and a broad variety of literature has been cited. The topic is timely and the findings appear original to me. Before I can strongly endorse publication, I would ask the authors for the following clarification, though:

1. It is not entirely clear to me what we learn from the time-resolved XRD data. It seems that the different excitation and probing depths in this method make the data hard to interpret. More importantly, there seems to be a slight disconnect between the results in Fig. 2h and the claim (page 8): "In this out-of-equilibrium regime, the average displacement of the ionic cores around their mean positions ... is small, as the nonthermal phonon population is dominated by high-frequency, low-amplitude optical phonons." Does the strong change of the trXRD peak in Fig. 2h not suggest the opposite? Could the authors clarify, please? If there is no absolute need for the trXRD data, would it make sense to move the trXRD data into the supplement?

2. To extract an effective temperature, the authors fit a Fermi Dirac distribution to their ARPES data. This procedure presumably implies that the electrons interact only weakly. Could the authors comment in which way strong correlations might cause a similarly broad energy distribution even for much colder electron systems? On the same note, it would be very helpful if the transient data in Fig. 2b showed horizontal and vertical error bars. This is important because on page 5, the authors broadly claim that the CDW order emerges at " $T_e \approx 2T_c$ ". Gazing at Fig. 2b, I would have estimated that the curve significantly peaks out of the noise floor only at ~ 490 K.

3. A minor recommendation: The agreement between experiment and theory in Fig. 2f is very good – especially given the complexity of the system. I would discourage overclaiming an "excellent agreement" (page 7), though. Quite understandably, the absolute oscillation frequencies do not perfectly agree. I would even go as far as saying that I would be suspicious if they perfectly agreed, in light of the complexity of the dynamics under discussion.

Response letter – Manuscript NCOMMS-20-43887

Nonequilibrium Charge-Density-Wave Order Beyond the Thermal Limit

January 22, 2021

We thank all referees for their careful reading of the manuscript and their valuable comments. Following the reviewers' advice, we hope we could clarify various aspects in our revision and address all concerns. Below we give a detailed point by point reply to each of the referees' comments. Modifications to the original manuscript have been marked in red.

Referee report 1:

The authors present a very interesting study on the non-equilibrium stabilization of a symmetry broken phase due to the transient suppression of low-frequency lattice fluctuations. I am particularly captivated by the extremely convincing application of the Ginzburg-Landau theory for the quantitative description of the experimentally observed non-equilibrium dynamics. The concept is novel and is obviously capable of providing a comprehensive and consistent interpretation of nonequilibrium processes associated with the destruction and formation of broken symmetry states of matter. In my opinion, it has the potential to become in the future a standard tool for the analysis and interpretation of experimental data in this research field. The presented experimental data is of very high quality and the analysis of the data has been performed with care. The interpretation of the result is sound and strongly supported by the simulations (see my comments above). Overall I strongly recommend the manuscript for publication in Natures Communications.

We would like to thank Reviewer 1 for his/her positive comments related to our work.

However, before a final decision the authors should address the following points: The authors introduce the inverted in-gap intensity at EF as a metric for the electronic order parameter because of practical limitations in determining the energy gap. I am wondering how changes in the thermal population of states near EF affect this metric as the electron temperature increases. I suspect that these effects are small at sufficiently low temperatures, i.e. under thermal equilibrium conditions. However, in the experiments transient electron temperatures of up to 2300 K are observed.

The inverted in-gap intensity is derived from a region of interest approximately symmetric around the Fermi-level, see box 1 in Fig. 2b. Therefore, no pronounced dependence from the electronic temperature is expected. This is further confirmed by the direct correspondence between the inverted in-gap intensity and the shadow-band intensity over the total fluence range and for early pump-probe delays (corresponding to high electronic temperatures). We now clarify this in the main text, p. 3 : *'We find that this metric – for the chosen region of interest and our experimental resolution – follows a BCS-like temperature dependence in equilibrium, as confirmed by static measurements (black markers in Fig. 3b), and thus is considered equivalent to $|\psi_e|$.* *Further, as the inverted in-gap intensity is derived from a region where the gap is centered*

around E_F , it is unaffected by thermal changes to the distribution function. ...

The authors mention a CDW melting threshold of 0.1 mJ/cm². It is not obvious to me, how this value was determined from the experimental data. This point should be clarified in a revised version.

We thank the referee for pointing out this ambiguity. Indeed, the trARPES data suggests a threshold fluence close to 0.05 mJ/cm². At this threshold fluence, the electronic CDW signatures, namely the energy gap and shadow band intensity, vanish almost completely (in contrast to the low-fluence amplitude mode regime, at which these CDW features are modulated only weakly). A second feature at the threshold fluence is the slowing-down of the CDW melting time, which also occurs near $F = 0.05$ mJ/cm² (see Fig. S7c).

We adapted the value of the threshold fluence in the text, and clarify the characteristic features in more detail (results, p. 3): *‘For a low absorbed fluence of 0.025 mJ/cm² below the CDW melting threshold, we observe a weak modulation of the CDW gap and SB intensity corresponding to the AM of the CDW at $\omega_{AM}/2\pi = 2.2$ THz (see Supplementary Fig. 3). At the CDW melting threshold ≈ 0.05 mJ/cm², the AM softens and becomes overdamped, while the CDW melting time t_{melt} slows down, and the energy gap and SB intensity vanish almost completely.’*

They authors furthermore write that “at the melting threshold ... the CDW melting time t_{melt} slows down”. Actually, Fig S7 implies that the slowing down of t_{melt} (as determined from the trARPES data) stops already way below the CDW melting threshold (at approx. 0.05 mJ/cm²) and that this conclusion is based only on a single data point (which makes this conclusion somehow questionable if only based on the trARPES data). Notably, the trXRD data (Fig. 2f) confirms a slowing down of t_{melt} up to a fluence of approx. of 0.1 mJ/cm². So there seems to be some mismatch between the trARPES data and the trXRD data. I suspect that this difference arises from the difference in the probing depth of the two techniques (see second paragraph on page 8). Overall, I feel that this part shows in the present form some inconsistencies. I therefore suggest to revise this paragraph correspondingly.

As discussed in the comment above, we adapted the fluence value of the melting threshold, which is now consistent with the statement mentioned by the referee (“at the melting threshold ... the CDW melting time t_{melt} slows down”). While we do not investigate the critical slowing-down in the low-fluence regime in great detail, our observations match a recent study of the related compound LaTe₃ [1], which focuses on the dynamical slowing-down using several complementary techniques. Further, our theoretical model is in good agreement with the experimental melting times over the full fluence range (Supp. Fig. 7c). Therefore, we are convinced that the CDW melting times obtained from trARPES display such critical behaviour near the phase transition.

As already mentioned by the referee, the melting times extracted from the trXRD data have to be treated with caution due to the contribution of sub-surface layers with reduced excitation densities. Nevertheless, our layered model also yields a reasonable description of the CDW melting behaviour extracted with trXRD. In this context, the observation of a maximum quench of SL intensity at ≈ 0.1 mJ/cm² – slightly above the threshold fluence extracted from ARPES measurements – is expected, as higher fluences are required to melt the SL order in buried layers. We adapt a sentence discussing the XRD results (p. 5): *‘Furthermore, while the SL intensity I_{SL} drops linearly with excitation density shortly after excitation, this behaviour plateaus after crossing a fluence of the melting threshold of ≈ 0.1 mJ/cm²’*

The agreement between experimental data and simulations based on the Landau-Ginzburg model is indeed excellent, particularly as only two fitting parameters are used. One notices some differences in the 1 to 10 ps regime, where the experimental data show a much smoother recovery than the simulations (here, the loga-

rhythmic presentation may be a little bit misleading). Could the authors comment on the origin of the more step-like recovery in the simulation. Is there any obvious reason, why the match between experiment and simulations improves as the fluence increases?

The delayed restoration of CDW order present in the model (Fig. 3a) is connected to a dynamical slowing-down of the recovery, briefly discussed in Supp. Inf. E: As sketched in Fig. S8, after few damped initial oscillations around the potential minimum of the high-symmetry potential, the order parameter $|\psi|$ and its velocity $\delta\psi/\delta t$ are close to 0. At the same time, the potential recovers its ground-state double-well shape ($\eta = 1$, see purple line). However, the slope of the potential in proximity to 0, which defines the kinetics of $|\psi|$ during the recovery at ≈ 2 ps, is rather flat. Thus, the order parameter does not directly follow the potential minimum, but gets trapped at the metastable state at 0, corresponding to the slightly delayed recovery in our model. However, once $|\psi|$ leaves this metastable position, the system quickly relaxes to the global potential minimum, apparent from the fast ‘step-like’ recovery. Similar to a dynamical slowing-down of the CDW melting, this slowdown of the recovery is mitigated in real systems (due to coupling to other modes, crystal defects, CDW nucleation processes and topological defects, an inhomogeneous excitation profile and coupling to sub-surface layers), which, however, goes beyond our current minimal model.

We added a brief discussion of this additional critical behaviour to the main text, tdGL theory (p. 8): ‘Close to this metastable point, the potential is rather flat, leading to a critical slowing-down of the order-parameter dynamics [1], discussed in detail in Supplementary Note E. *A similar critical behaviour is expected during the recovery of the CDW order. In the overshoot regime, after dampening of the initial oscillations, the order parameter can get trapped at the metastable local maximum despite an incipient recovery of the double-well ground state. However, in real systems, several microscopic processes, such as local modification of T_C by crystal defects [2, 3], CDW nucleation and creation of topological defects [4] and coupling of the collective excitation to other phonons [5], will screen against a pronounced critical slowing-down...*

Further, in the discussion of the critical slowing-down during the recovery in Supp. Inf. E, we changed the wording for improved clarity, and now link to Fig. 3a: ‘... *Due to the weak curvature in the vicinity of $|\psi| = 0$, the system is trapped in a metallic phase, despite an emerging double-well potential. However, this divergence is difficult to observe experimentally, as it occurs at narrow fluence windows, and is, similar to the slowing-down of the CDW melting, suppressed by crystal defects, coupling to other phonon modes and an inhomogeneous excitation profile. This critical behaviour leads to a delayed onset of CDW recovery in the simulations as compared to the electronic CDW dynamics for certain fluences, see Fig. 2f (curve $F = 0.1$ mJ/cm²) and Fig. 3a.*

Further, we deliberately chose not to fit the recovery timescale, but to use lattice relaxation time scales reported in literature for a similar material in order to keep the model as simple as possible. This neglects a potential fluence dependence of the underlying lattice thermalization processes, which might be a further contribution to the observed discrepancy at intermediate fluences.

Based on the introduced model, is it possible to somehow extrapolate to an initial lattice temperature $T_l = 0$ K? This could provide valuable information on how much lattice fluctuations contribute to a reduction of T_C in comparison to T_{MF} .

We would like to thank the referee for pointing out this interesting idea. However, to perform a meaningful extrapolation of T_C at $T_l = 0$ K, the fluence-dependent analysis of the electronic CDW gap and SL reflections would need to be performed at various sample temperatures. Unfortunately, the trARPES and trXRD measurements have been performed only at a single temperature (100 K), and a detailed mapping of

the multi-dimensional parameter space ($I_{\text{in-gap}}$ and I_{SL} as functions of pump-probe delay, fluence and initial temperature) is extremely challenging.

Referee report 2:

The authors combine time-resolved ARPES with time-resolved x-ray diffraction for studying charge density wave dynamics in TbTe_3 . They demonstrate nonthermal CDW order, existing far above the critical temperature measured in equilibrium. With tr-ARPES, they show how the CDW order recovers at temperatures higher than the equilibrium critical temperature. For explaining the experimental observations, the authors propose a phenomenological description where the transition temperature, out-of-equilibrium, changes transiently. While the tr-ARPES data is of superb quality, the modeling unfortunately only poorly reproduces the experimental observations. The proposed time-dependent, non-equilibrium critical temperature is an interesting idea. Yet, the choice of the lattice temperature as the origin is insufficiently explained, and the underpinning mechanisms remain unclear. Of course, the critical temperature is known to change with pressure, doping, or disorder – all phenomena that rival the newly proposed mechanism. The recovery timescales predicted by the model disagree with the data by a factor of two in Figure 3a. This disagreement hints at an underlying flaw in the model.

We thank the referee for his/her comments. While we agree that the simulations do not perfectly quantitatively reproduce the experimental curves at all fluences, one has to consider the complexity of the system, the large range of applied fluences and the different experimental methods. To describe the data, we apply a **minimal** model with only 2 (global) free parameters, namely the damping and the nonthermal critical temperature, to describe 20 curves of structural and electronic CDW features in a regime of weak perturbation up to a regime of full CDW melting and recovery. As recognized by referees 1 and 3, it is quite remarkable that we are able to reproduce all main features in the trARPES and the trXRD datasets with such a simple model and a limited number of free parameters. Therefore, we strongly disagree with the notion that our model is conceptually flawed. Improving this model by including coupling to other modes, a separate treatment of the structural and electronic degrees of freedom, treatment of CDW nucleation processes and topological defects may yield an even better description in the future. However, our minimal model provides a fundamental understanding of the collective CDW excitations over a large range of fluences and of the underlying potential energy surface. To emphasize this point, we adapted the wording in the section tdGL theory (p. 7), see also our reply to the final remark of referee 3: *‘Given the complexity of the system, this model with its minimal amount of free parameters is in remarkable agreement with the electronic order parameter...’*

While the critical temperature is known to change with pressure [6, 7] and disorder [3, 8], both effects would lead to a **lower** critical temperature with increasing pressure or disorder such as could be expected after excitation, not an enhancement. Further, as we track the position of the Fermi level in ARPES, we can exclude a significant variation of doping. Additionally, if a transient modification of pressure, disorder or doping due to the optical excitation would lead to a transient increase in critical temperature, this should feature a clear fluence dependence. However, we do not observe a strong fluence dependence of the nonthermal critical temperature – in agreement with our proposed model. Thus, a transient change of pressure, doping or disorder can not explain the observed nonthermal enhancement. Additionally, our proposed scenario provides a *microscopic* explanation for the transiently enhanced critical temperature, which is much more powerful and predictive compared to a scenario via a modification of a *macroscopic* property, as suggested by the referee.

Regarding the criticism of the proposed model, it is unclear where exactly the lack of clarity of the underpinning mechanism lies, as the claim of an insufficient explanation is not further substantiated. We attribute

the observed CDW formation at increased electronic temperatures to a reduction of CDW-inhibitory lattice fluctuations in the out-of-equilibrium scenario of hot electrons and a cold lattice, which is recognized by referees 1 and 3 as physically sound and convincing.

The mismatch of the recovery timescales in the model mentioned by the referee (and also addressed by ref. 1) is mostly related to the critical slowdown in the recovery observed in the model, and discussed in the Suppl. Sec. E. (see reply to point d and reply to ref. 1). Additionally, we want to point out that we deliberately chose not to fit the recovery timescale, but to use lattice relaxation time scales reported in literature for a similar material in order to keep the model as simple as possible. This neglects a potential fluence dependence of the underlying lattice thermalization processes, and neglects further details of the CDW recovery (CDW domain formation, topological defects). This might also explain some of the discrepancy in the recovery, but in our opinion increases the illustrative power of the model by keeping it as simple as possible.

This paper's central point is that the CDW recovers at electronic temperatures above the critical temperature when away from equilibrium. I believe, to convince the reader that this is indeed the point, Figure 3 deserves a revision. a. In Fig. 4b, at high temperatures, the inverted in-gap intensity is as high as 0.3. Is this the uncertainty of the measurement (similar scatter is visible in Fig S2)? If yes, then all values above 400 K are within uncertainty consistent with the equilibrium curve.

The 'scatter' at high temperatures mentioned by the referee in Fig. 3b and Fig. S2 is not related to the uncertainty of the measurement, but corresponds to the initial collective CDW oscillations, and is labelled as such in Fig. 3b (we believe the referee refers to this figure, and not to Fig. 4b). To emphasize this point, we made a minor change to the arrow that marks the oscillation regime. Further, to highlight the deviation from the equilibrium curve, we added error bars to the transient curve in Fig. 3b. By including the uncertainty, it becomes obvious that the recovery of CDW order already sets in at ≈ 600 K, significantly above thermal $T_c = 336$ K, see the revised Fig. 3 below. We adapt a sentence in the results, p. 5: '*...In the out-of-equilibrium setting, CDW order reappears below $T_c \approx 600$ K (yellow shaded area), indicating an increased effective critical temperature T_c^* ...*'

b. It is hard to see what colors represent what time delays in Fig. 3b.

For improved recognition of the delay values, we added minor ticks to the color legend in Fig. 3b.

c. In Figure 3b, the BCS curve seems to be distinctly different from the data for temperatures below 200 K. Do the authors have an explanation why their new method of determining the bandgap deviates from BCS?

A similar deviation of the CDW gap size from BCS behavior has already been observed previously, see for example Fig. 3h in Moore et al. for static ARPES experiments of a related tritelluride [9]. The slope of our static temperature series qualitatively agrees with the curve of Moore et al., featuring a flatter slope than the BCS curve at intermediate temperatures. Therefore, our analysis yields comparable results to a more systematic extraction of the CDW gap size.

d. In Fig. 3a, the dashed line reproduces the experimental data well for large fluences; however, the recovery time appears to differ by a factor of two for lower fluence.

As mentioned above, we chose not to fit the recovery timescale, but to use lattice relaxation time scales reported in literature for a similar material in order to keep the model as simple as possible. This neglects

Revised Fig. 3. CDW recovery dynamics. (a) Time evolution of the inverted in-gap intensity in the high-fluence regime (displaced vertically). Normalized time-dependent Ginzburg-Landau simulations are shown in black. (b) Inverted in-gap intensity versus extracted electronic temperatures. **One standard deviation of the T_e fit (horizontal error bars) and one standard deviation derived from electron counting statistics (vertical error bars) are given as uncertainty.** $\tilde{I}_{\text{in-gap}}$ extracted from a static temperature series (black markers, T -values from heater setpoints, curve normalized to the lowest accessible T -value) is in general agreement with the BCS-type T -dependence of the order parameter (grey curve). The dynamic trace shows the full cycle of laser-heating and CDW melting, coherent oscillations and CDW recovery (delay encoded in the color code). The yellow shaded area marks the region of dynamical CDW formation at electronic temperatures above T_c . The pre-excitation value of the dynamic trace ($T = 100$ K) is normalized to the corresponding value of the static T -dependence.

a potential fluence dependence of the underlying lattice thermalization processes, which might lead to the observed discrepancy. A further effect is the dynamical slowing-down of the recovery, which is briefly discussed in Supp. Inf. E. This effect can lead to a delayed restoration of CDW order in the simulations. (Partially citing our reply to referee 1:) As sketched in Fig. S8, after few damped initial oscillations around the potential minimum of the high-symmetry potential, the order parameter $|\psi|$ and its velocity $\delta\psi/\delta t$ are close to 0. At the same time, the potential recovers its ground-state double-well shape ($\eta = 1$, see purple line). However, the slope of the potential in proximity to 0, which defines the kinetics of $|\psi|$ during the recovery at ≈ 2 ps, is rather flat. Thus, the order parameter does not directly follow the potential minimum, but gets trapped at the metastable state at 0, corresponding to the slightly delayed recovery in our model. Similar to a dynamical slowing-down of the CDW melting, this slowdown of the recovery is mitigated in real systems (due to coupling to other modes, crystal defects, CDW nucleation processes and topological defects, an inhomogeneous excitation profile and coupling to sub-surface layers), which, however, goes beyond our current minimal model.

It is somewhat unclear what the tr-X-ray data contributes to the discovery. It is mentioned but not implemented into the discussion beyond that it is qualitatively comparable to tr-ARPES.

Firstly, while the interpretation of the trXRD data is less straightforward, we qualitatively show that within

the first picoseconds the structural CDW features closely follow the evolution of the electronic features: AM mode behaviour and a fast recovery of the CDW order after weak excitation and a strong persisting suppression of CDW order after strong excitation. Within our temporal resolution, we can thus describe the system using a single order parameter. We now emphasize this point in the results (p. 5): ‘...*Nonetheless, the trXRD data clearly shows that not only the electronic, but also the lattice superstructure is melted upon strong photoexcitation. The qualitative agreement of the electronic and structural response demonstrates a strong coupling between electronic and lattice degrees of freedom on ultrafast timescales, and suggests an equivalent treatment of $|\psi_s|$ and $|\psi_e|$ within the experimental time resolution...*’

The lattice information allows us to exclude another conceivable scenario, in which only the electronic CDW order is destroyed while the lattice distortion is not, which could lead to a similar nonthermal enhancement of the electronic critical temperature. A long-lived state of melted charge order with a persisting periodic lattice distortion was recently speculated in photoexcited 2H-NbSe₂ [10]. We try to clarify the benefit of the structural information in the first paragraph of the discussion p. 8: ‘*We unambiguously demonstrate a transient CDW behaviour distinct from equilibrium, as evidenced by the CDW AM modulations after weak excitation despite electronic temperatures exceeding thermal T_c , and from the CDW recovery at elevated electronic temperatures after strong excitation. The qualitative correspondence of charge and structural features of the CDW excludes a scenario in which only the electronic superstructure is destroyed while the lattice distortion remains intact, which could facilitate such a nonthermal behaviour. So what causes this enhanced transient stability of CDW order far beyond the equilibrium T_c ? ...*’

Secondly, despite the difference of techniques, fluence calibrations and complexity of the trXRD data, we achieve a respectable description of the SL intensities (Fig. 2h) by implementing a layered tdGL description, using the parameters of the trARPES fit (Fig. 2f) without introducing any additional free parameters in the layered model. This underlines the reliability of the tdGL model and the close correspondence of electronic and structural CDW features.

How did the authors calculate the absorbed fluence for both experiments?

The incident power was measured close to the sample position with a conventional power meter. The absorbed fluence was calculated using the incident power, the pump-laser repetition rate and spot size (see methods), and the complex refractive index of TbTe₃ ($n=0.9$ and $k=2.6$ at $\lambda=800$ nm, see methods).

The pulse durations in both measurements appear to be significantly different. 35 fs in tr-ARPES and 110 fs in time-resolved XRD. Since the authors observe dynamics on a 100 fs time scale, how does the difference modify the results?

The lower time resolution in trXRD smears out the coherent oscillations of the structural distortion and prevents their clear observation. Improved temporal resolution has allowed Trigo et al. to observe strongly fluence-dependent coherent modulations of the superlattice peak intensity in a closely related compound. In our model description, we account for the limited temporal resolution by convolution with a Gaussian, which leads to a respectable agreement with the experimental data, see Fig. 2h. As mentioned in the manuscript, see results on p. 5: ‘*In contrast to the electronic response, we do not observe clear coherent oscillations of the SL peak intensity upon strong excitation. This originates most likely from the lower temporal resolution of the trXRD setup and the contribution of sub-surface crystal layers with varying, lower excitation densities (see Supplementary Note D).*’

The Landau model fit in Figure 2f is well for intermediate fluences; however, it significantly deviates from

the data for low and high fluences. Also, in Figure 2h, the model fits substantially differ from the data, well outside the depicted uncertainties. The manuscript deserves a discussion on the discrepancy between the model and the data.

As mentioned above, we model a total of 20 curves over strongly varying excitation conditions (weak to strong perturbation including full CDW melting and recovery) and different experimental methods with a minimal model with only 2 global fit parameters. While the simulations do not match perfectly, we capture all relevant experimental features and trends. Further, we discuss discrepancies in several sections of our manuscript. For instance, a detailed description of the effects of the critical slowing-down (see main text, last 2 paragraphs of the tdGL framework (p. 8) and Supp. Inf. E) as well as a detailed record of uncertainties of the layered model (see Supp. Inf. D, discussion of fluence calibration and the penetration depths) can be found. Both referee 1 and 3 are convinced by the agreement of our simulations with the experimental curves, and referee 3 ‘would even go as far as saying that [he] would be suspicious if they perfectly agreed, in light of the complexity of the dynamics under discussion.’

The deviation for a low fluence of 0.05 mJ/cm^2 in Fig. 2f can be explained by a critical slowing-down near the threshold fluence, which is more pronounced in the simulations. In the high-fluence regime, the simulations describe the initial modulation frequency and the recovery quite well. However, the absolute oscillation amplitude is overestimated. While including, e.g., a more complex and fluence-dependent damping term would improve the fit, we believe that these minor deviations are acceptable in favour of keeping the underlying model as simple as possible.

In Fig. 2h, the authors display photon counting statistics. What about FEL jitter in position, angle, and wavelength? Are these uncertainties negligible?

The trXRD measurements were performed at a synchrotron slicing source (FEMTO hard X-ray slicing source (X05LA), Swiss Light Source, Paul Scherrer Institut). The utilized femtosecond X-ray pulses behave similar to synchrotron radiation, and do not feature any relevant jitter in position, angle or wavelength. Therefore, the major source for uncertainty results from counting statistics. We now clarify this further in the methods section, p. 10 :*‘The utilized laser-sliced X-ray pulses ($h\nu_{\text{X-ray}}=7 \text{ keV}$, $\Delta t \approx 120 \text{ fs}$) feature the high stability of conventional synchrotron radiation and do not exhibit any relevant jitter in position, angle or wavelength. The diffracted X-ray intensity was recorded with an avalanche photodiode in an asymmetric diffraction geometry...’*

Minor comments In Fig. 2 b, the indication for box 3 is missing.

We were not able to reproduce this bug of a missing indication in Fig. 2b with several pdf readers. We re-rendered the figure and hope this solves the problem.

In Figure 5 it is difficult to see the point. How would the picture look for an equilibrium CDW order state and how would that picture differ from the non-equilibrium CDW order state.

In this simple schematic, the nonequilibrium CDW state can be considered similar to the equilibrium CDW order, in which both lattice and charges exhibit a superstructure. The only difference between the equilibrium and non-equilibrium case is the elevated electronic temperature in the perturbed nonthermal scenario. To underline the concept of the nonthermal CDW order, we modified the schematic, added the equilibrium low-temperature CDW scenario requested by the referee, and adapted the color scheme and the figure caption, see the revised version of Fig. 5 below.

Revised Fig. 5. Illustration of nonthermal CDW order. In equilibrium at elevated temperatures, the system is in a trivial metallic phase (top). The charge density (wavy line) and the mean positions of the ionic cores (circles) are spaced evenly, as strong thermal lattice fluctuations prevent long-range CDW order. In equilibrium at low temperatures, the system features an ordered charge- and lattice superstructure (center). Photoexcitation of the CDW ground state ($T_{\text{pre-exc.}} \ll T_c$) generates a hot electron distribution, while the lattice initially remains cold (bottom). In this out-of-equilibrium state, thermal lattice fluctuations are weak and barely hinder long-range CDW ordering. Hence, the charge and lattice superstructure is stabilized at electronic temperatures beyond T_c .

Referee report 3:

An interesting time-resolved ARPES study of transient melting and recovery of charge density wave (CDW) order in TbTe_3 is presented. The authors follow the characteristic CDW-related energy gap and shadow bands as a function of time after optical excitation and repeat the experiment for a broad variety of pump fluences. From these data they extract electronic quasi-temperatures as a function of delay time and fluence and find that, during the electronic cool down, the CDW gap opens already at electron temperatures distinctly above the equilibrium transition temperature T_c . They explain this scenario using the Ginzburg-Landau theory and a three-temperature model and suggest that after optical excitation the crystal lattice remains much cooler than T_c and, thus, phonon fluctuations do not destabilize the CDW order. Under these conditions, even rather hot electrons may form a CDW. The manuscript also features trXRD data taken under comparable conditions.

Overall, this is obviously a very carefully executed study which uses some of the most sophisticated, expensive and exclusive experimental techniques available at this time. The data have been treated with care and a broad variety of literature has been cited. The topic is timely and the findings appear original to me.

We would like to thank Reviewer 3 for his/her positive comments related to our work.

Before I can strongly endorse publication, I would ask the authors for the following clarification, though:
 1. It is not entirely clear to me what we learn from the time-resolved XRD data. It seems that the different excitation and probing depths in this method make the data hard to interpret. More importantly, there seems to be a slight disconnect between the results in Fig. 2h and the claim (page 8): "In this out-of-equilibrium regime, the average displacement of the ionic cores around their mean positions ... is small, as the nonther-

mal phonon population is dominated by high-frequency, low-amplitude optical phonons." Does the strong change of the trXRD peak in Fig. 2h not suggest the opposite? Could the authors clarify, please? If there is no absolute need for the trXRD data, would it make sense to move the trXRD data into the supplement?

(Partially citing our response to referee 2) Firstly, while the interpretation of the trXRD data is less straightforward, we qualitatively show that within the first picoseconds the structural CDW features closely follow the evolution of the electronic features: AM mode behaviour and a fast recovery of the CDW order after weak excitation and a strong persisting suppression of CDW order after strong excitation. Within our temporal resolution, we can thus describe the system using a single order parameter. We now emphasize this point in the results (p. 5): ‘...*Nonetheless, the trXRD data clearly shows that not only the electronic, but also the lattice superstructure is melted upon strong photoexcitation. The qualitative agreement of the electronic and structural response demonstrates a strong coupling between electronic and lattice degrees of freedom on ultrafast timescales, and suggests an equivalent treatment of $|\psi_s|$ and $|\psi_e|$ within the experimental time resolution...*

The lattice information allows us to exclude another conceivable scenario, in which only the electronic CDW order is destroyed while the lattice distortion is not, which could lead to a similar nonthermal enhancement of the electronic critical temperature. A long-lived state of melted charge order with a persisting periodic lattice distortion was recently speculated in photoexcited 2H-NbSe₂ [10]. We try to clarify the benefit of the structural information in the first paragraph of the discussion p. 8: ‘*We unambiguously demonstrate a transient CDW behaviour distinct from equilibrium, as evidenced by the CDW AM modulations after weak excitation despite electronic temperatures exceeding thermal T_c , and from the CDW recovery at elevated electronic temperatures after strong excitation. The qualitative correspondence of charge and structural features of the CDW excludes a scenario in which only the electronic superstructure is destroyed while the lattice distortion remains intact, which could facilitate such a nonthermal behaviour. So what causes this enhanced transient stability of CDW order far beyond the equilibrium T_c ? ...*

Secondly, despite the difference of techniques, fluence calibrations and complexity of the trXRD data, we achieve a respectable description of the SL intensities (Fig. 2h) by implementing a layered tdGL description, using the parameters of the trARPES fit (Fig. 2f) without introducing any additional free parameters in the layered model. This underlines the reliability of the tdGL model and the close correspondence of electronic and structural CDW features.

Finally, when considering changes of diffraction intensities, two transient effects need to be distinguished: (i) a change of the structure factor due to a change of the **mean** atomic positions, e.g. the atomic motions related to the suppression of the periodic lattice distortion. This leads to the strong suppression of the superlattice peaks, which have a vanishing structure factor in the high temperature phase. (ii) a variation of the Debye-Waller factor, corresponding to random fluctuations of the atoms around these mean positions. This effect leads to a slight suppression of the main lattice reflections, and transfer of intensity into the non-thermal background (and also a minor further suppression of the superlattice reflections) on the much slower timescale of lattice thermalization, see e.g. [11]. We clarify this point in the first paragraph of the results p. 2: ‘...*These SL peaks arise from the periodic lattice distortion associated with the CDW, and are displaced by the CDW wave vector $\pm q_{\text{CDW}}$ from the main peak positions [12, 13]. As Fig. 1c shows, photoexcitation strongly suppresses the SL peak corresponding to a rearrangement of the atomic mean positions towards the trivial metallic phase, while the main lattice peak reflecting the average crystal structure shows only minor changes.*

2. To extract an effective temperature, the authors fit a Fermi Dirac distribution to their ARPES data. This

procedure presumably implies that the electrons interact only weakly. Could the authors comment in which way strong correlations might cause a similarly broad energy distribution even for much colder electron systems? On the same note, it would be very helpful if the transient data in Fig. 2b showed horizontal and vertical error bars. This is important because on page 5, the authors broadly claim that the CDW order emerges at " $T_e \approx 2T_c$ ". Gazing at Fig. 2b, I would have estimated that the curve significantly peaks out of the noise floor only at ≈ 490 K.

The discussed Fermi-Dirac fitting procedure is a standard method in ARPES, and has been successfully applied also to strongly correlated materials, such as cuprates [14]. Indeed, several effects can complicate a quantitative determination of T_e , such as a complex band dispersion in proximity to the Fermi-cutoff and many-body interactions (e.g. due to the formation of a quasiparticle peak close to the Fermi level or due to transfer of spectral weight away from the Fermi level by Mott physics). However, firstly, electron-electron interactions are relatively weak in the tritellurides. Secondly, for the Fermi-Dirac fitting procedure, we choose an energy-momentum region that features a sharp metallic band with a rather linear dispersion at the Fermi level (see right side of Fig. 2b-c). Thirdly, apart from the non-thermalized regime during pump-probe overlap, the Fermi-Dirac fits provide an excellent description of the experimental energy distribution curves (Fig. S5a).

As requested by the referee, we added error bars to Fig. 3b (we believe the referee is referring to this figure, and not to Fig. 2b), see our reply to referee 2 for the revised figure. The horizontal error bars correspond to one standard deviation of the T_e Fermi-Dirac fits. The vertical error bars are derived from electron counting statistics by calibrating the number of detected events on the CCD camera per electron, which allows us to estimate the total number of detected electrons in a certain region of interest (To prevent any clutter due to the error bars and high density of data points in the dynamic trace, for delays > 10 ps only every third data point is shown in the figure). As the uncertainty in the relevant recovery regime is on the order of ± 50 K and ± 0.05 in vertical direction, it becomes more obvious that the recovery of CDW order already sets in at ≈ 600 K. We adapt the sentence mentioned by the referee accordingly (results, p. 5): ‘...In the out-of-equilibrium setting, CDW order *reappears below $T_e \approx 600$ K* (yellow shaded area), indicating an increased effective critical temperature T_c^*’

Further, a description of the error bars has been added to the caption of Fig. 3b: ‘*One standard deviation of the T_e fit (horizontal error bars) and one standard deviation derived from electron counting statistics (vertical error bars) are given as uncertainty.*’

3. A minor recommendation: The agreement between experiment and theory in Fig. 2f is very good – especially given the complexity of the system. I would discourage overclaiming an "excellent agreement" (page7), though. Quite understandably, the absolute oscillation frequencies do not perfectly agree. I would even go as far as saying that I would be suspicious if they perfectly agreed, in light of the complexity of the dynamics under discussion.

We agree with the referee and adapted the wording in the section tdGL theory (p. 7) accordingly: ‘*Given the complexity of the system, this model with its minimal amount of free parameters is in remarkable agreement with the electronic order parameter...*’

Further changes:

- Added vertical error bars to Fig. S3 derived from electron counting statistics, as discussed in the reply to referee 3, and modified the figure caption accordingly.

References

- [1] Alfred Zong, Pavel E Dolgirev, Anshul Kogar, Emre Ergeçen, Mehmet B Yilmaz, Ya-Qing Bie, Timm Rohwer, I-Cheng Tung, Joshua Straquadine, Xirui Wang, et al. Dynamical slowing-down in an ultrafast photoinduced phase transition. Physical review letters, 123(9):097601, 2019.
- [2] CJ Arguello, SP Chockalingam, EP Rosenthal, L Zhao, C Gutiérrez, JH Kang, WC Chung, RM Fernandes, S Jia, AJ Millis, et al. Visualizing the charge density wave transition in 2H-NbSe₂ in real space. Physical Review B, 89(23):235115, 2014.
- [3] Alan Fang, Joshua AW Straquadine, Ian R Fisher, Steven A Kivelson, and Aharon Kapitulnik. Disorder-induced suppression of charge density wave order: STM study of Pd-intercalated ErTe₃. Physical Review B, 100(23):235446, 2019.
- [4] Alfred Zong, Anshul Kogar, Ya-Qing Bie, Timm Rohwer, Changmin Lee, Edoardo Baldini, Emre Ergeçen, Mehmet B Yilmaz, Byron Freelon, Edbert J Sie, et al. Evidence for topological defects in a photoinduced phase transition. Nature Physics, 15(1):27–31, 2019.
- [5] RV Yusupov, T Mertelj, J-H Chu, IR Fisher, and D Mihailovic. Single-particle and collective mode couplings associated with 1-and 2-directional electronic ordering in metallic RTe₃ (R= Ho, Dy, Tb). Physical review letters, 101(24):246402, 2008.
- [6] A. Sacchetti, E. Arcangeletti, A. Perucchi, L. Baldassarre, P. Postorino, S. Lupi, N. Ru, I. R. Fisher, and L. Degiorgi. Pressure dependence of the charge-density-wave gap in rare-earth tritellurides. Phys. Rev. Lett., 98:026401, Jan 2007.
- [7] DA Zocco, JJ Hamlin, K Grube, J-H Chu, H-H Kuo, IR Fisher, and MB Maple. Pressure dependence of the charge-density-wave and superconducting states in gdte 3, tbte 3, and dyte 3. Physical Review B, 91(20):205114, 2015.
- [8] JAW Straquadine, F Weber, S Rosenkranz, AH Said, and IR Fisher. Suppression of charge density wave order by disorder in pd-intercalated erte 3. Physical Review B, 99(23):235138, 2019.
- [9] RG Moore, V Brouet, R He, DH Lu, N Ru, J-H Chu, IR Fisher, and Z-X Shen. Fermi surface evolution across multiple charge density wave transitions in ErTe₃. Physical Review B, 81(7):073102, 2010.
- [10] Daniel T Payne, Paolo Barone, Lara Benfatto, Fulvio Parmigiani, and Federico Cilento. Lattice contribution to the unconventional charge density wave transition in 2h-nbse _2: a non-equilibrium optical approach. arXiv preprint arXiv:2010.09826, 2020.
- [11] Pavel E Dolgirev, AV Rozhkov, Alfred Zong, Anshul Kogar, Nuh Gedik, and Boris V Fine. Amplitude dynamics of the charge density wave in LaTe₃: Theoretical description of pump-probe experiments. Physical Review B, 101(5):054203, 2020.
- [12] AW Overhauser. Observability of charge-density waves by neutron diffraction. Physical Review B, 3(10):3173, 1971.

- [13] N Ru, CL Condon, GY Margulis, KY Shin, J Laverock, SB Dugdale, MF Toney, and IR Fisher. Effect of chemical pressure on the charge density wave transition in rare-earth tritellurides $R\text{Te}_3$. Physical Review B, 77(3):035114, 2008.
- [14] S Parham, H Li, TJ Nummy, JA Waugh, XQ Zhou, J Griffith, J Schneeloch, RD Zhong, GD Gu, and DS Dessau. Ultrafast gap dynamics and electronic interactions in a photoexcited cuprate superconductor. Physical Review X, 7(4):041013, 2017.

REVIEWER COMMENTS

Reviewer #1 (Remarks to the Author):

The authors replied to the reviewers comments in a very comprehensive and satisfying manner. I now stongly recommend publiaction of the manscript in Nature Communication.

Reviewer #2 (Remarks to the Author):

In the revised manuscript, the authors appropriately responded to most comments. Nevertheless, in my opinion, the validity of the model still deserves a more detailed discussion. In response to my comments, the authors reply, "a microscopic explanation for the transiently enhanced critical temperature [] is much more powerful and predictive compared to a scenario via a modification of a macroscopic property." I agree that a microscopic model is more powerful. My refined question is then: what is the microscopic mechanism that raises the transition temperature?

The authors attribute the discrepancy between the model and the data to the complexity of the system. I understand the system is complex, and the attempt to describe it with a simple model is laudable. Nevertheless, because of the disagreement, it is difficult to judge how significant the introduction of the transient transition temperature is to the model. Does the model reproduce the data similarly well if the transition temperature is kept constant? Is the choice of $\tau_{\text{ph-ph}}$ justified? (in figure 3a, the model predicts a consistently lower order-parameter than the experiment for 1-5 ps, exactly where the transition temperature measured in ARPES is above the equilibrium transition temperature).

I would like to reiterate that the time-resolved ARPES data appears of the highest quality, is well presented, and the experimental observation of the increased transition temperature out-of-equilibrium is novel and exciting. In my opinion, the lack of microscopic insights into the model remains underwhelming.

Reviewer #3 (Remarks to the Author):

The authors have extensively replied to all questions and made occasional adaptations in the manuscript. I find the scarce changes still acceptable and I support publication of the work as is.

Response letter 2 – Manuscript NCOMMS-20-43887A

Nonequilibrium Charge-Density-Wave Order Beyond the Thermal Limit

February 27, 2021

We thank all referees for their careful reading of the revised manuscript. Below we give a point by point reply to the remaining concerns. Modifications to the manuscript have been marked in red.

Referee report 1:

The authors replied to the reviewers comments in a very comprehensive and satisfying manner. I now strongly recommend publication of the manuscript in Nature Communication.

We thank the referee for his/her positive comments.

Referee report 2:

In the revised manuscript, the authors appropriately responded to most comments. Nevertheless, in my opinion, the validity of the model still deserves a more detailed discussion. In response to my comments, the authors reply, “a microscopic explanation for the transiently enhanced critical temperature [] is much more powerful and predictive compared to a scenario via a modification of a macroscopic property.” I agree that a microscopic model is more powerful. My refined question is then: what is the microscopic mechanism that raises the transition temperature?

We thank the referee for pointing out remaining ambiguities regarding the microscopic mechanism. We further extended the discussion in the main manuscript:

‘In equilibrium, lattice fluctuations induced by thermally populated phonons, accompanied by fluctuations of the charge density, reduce T_c significantly below the mean-field value T_{MF} ’. In other words, in thermal equilibrium of electrons and lattice, incoherent lattice fluctuations lead to a suppression of long-range 3D CDW order above the critical temperature T_c .

In contrast, ‘optical perturbation breaks the thermal equilibrium between charges and lattice. Initially, electrons and certain optical phonons are strongly excited, while the overall vibrational population of the lattice – determined by acoustic modes that account for the majority of the lattice heat capacity – is still close to its pre-excitation value corresponding to an effective lattice temperature significantly below T_c . In this out-of-equilibrium regime, the average displacement of the ionic cores around their mean positions (mean-squared displacement) is small, as the nonthermal phonon population is dominated by high-frequency, low-amplitude optical phonons [1]. Thus, initially after excitation, lattice fluctuations are strongly suppressed and counteract a mean-field long-range ordering only weakly, which facilitates CDW formation even at electronic temperatures far beyond T_c , illustrated in Fig. 5.’ In other words, the atomic mean-squared displacement is small in the nonthermal regime after excitation and thus long-range 3D CDW order is only weakly obstructed compared to a scenario in which lattice and electrons are in thermal equilibrium at a temperature

above T_c (see schematic Fig. 5).

‘Over the course of several ps, depending on the lattice thermalization time τ_{ph-ph} , energy is transferred from the strongly coupled optical hot phonons to the remaining phonon modes. This defines the crossover from the nonthermal to the quasi-thermal regime, at which electrons and lattice locally reach thermal equilibrium. As the lattice temperature rises, acoustic (high-amplitude) fluctuations and CDW phase fluctuations increase, which impedes long-range 3D CDW order, and T_c^ consequently converges towards the equilibrium T_c .’* In other words, on the timescale of energy transfer from optical to acoustic phonon modes, i.e. the lattice thermalization timescale τ_{ph-ph} , the CDW-inhibitory fluctuations increase.

The authors attribute the discrepancy between the model and the data to the complexity of the system. I understand the system is complex, and the attempt to describe it with a simple model is laudable. Nevertheless, because of the disagreement, it is difficult to judge how significant the introduction of the transient transition temperature is to the model. Does the model reproduce the data similarly well if the transition temperature is kept constant? Is the choice of τ_{ph-ph} justified? (in figure 3a, the model predicts a consistently lower order-parameter than the experiment for 1-5 ps, exactly where the transition temperature measured in ARPES is above the equilibrium transition temperature).

Using the time-dependent Ginzburg-Landau ansatz to simulate the data necessitates a strongly increased critical temperature at early times to capture the CDW melting time t_{melt} and initial oscillation frequencies. A subsequent crossover of the increased critical temperature T_c^* towards the equilibrium value $T_{c,equl}$ is also required to reproduce the observed frequency chirp, the fast recovery (on the timescale of 1-10 ps) and the slow recovery at quasi-equilibrium on the order of 100 ps. Simulations using the constant equilibrium critical temperature ($T_c = 336$ K) lead to a strong deviation from the observed frequencies and melting times, as the slope of the underlying potential, determined by $\eta(t) = T_e/T_c$ is strongly overestimated, see panel a) in the figure below. Further, the simulated recovery only sets in once $T_e < T_c$, which does not match experimental observations, see panel b) in the Figure below and Fig. 3b of the main manuscript.

We now mention this point explicitly in section Time-dependent Ginzburg-Landau theory in the main manuscript: *‘To illustrate the necessity of a transiently enhanced T_c^* to describe the data, we perform tdGL simulations keeping the critical temperature fixed at the equilibrium value, which, however, leads to a severe deviation from the experimental oscillations and CDW recovery, see Supplementary Fig. S5.’*

Further, we include the figure shown below in the Supplementary Information and expand this discussion in the last paragraph of Supplementary Section ‘Details of the tdGL simulations of the electronic order’: *‘As illustrated in Fig. 3b, the transient CDW recovery strongly deviates from static (thermal) behaviour, which necessitates the introduction of a transiently increased critical temperature T_c^* in the tdGL simulations. To highlight the requirement of a transiently increased T_c^* , we perform additional simulations employing the constant equilibrium critical temperature T_c , while keeping the remaining parameters fixed as described above. As Supplementary Fig. S5 shows, this does not reproduce the experimental data, as (i) the simulated oscillation frequencies are strongly overestimated due to the increased slope of the underlying potential energy landscape (determined by $\eta(t)$) and (ii) the simulated recovery sets in only at $T_e < T_c$ – at a significant delay with respect to the experimental data.’*

To circumvent these deviations and motivated by the experimental observation of the pronounced CDW stability at elevated electronic temperatures, we introduce the transiently increased critical temperature. As timescale connecting the initial nonequilibrium regime with the thermal regime, we choose the phonon thermalization time of the system τ_{ph-ph} based on the proposed microscopic mechanism, which correctly

reproduces the timescale of the fast recovery. The remaining deviations in the recovery process can be explained by additional contributions from secondary processes (critical slowing-down of the recovery) and the approximation of a (fluence-independent) τ_{ph-ph} extracted from a related material, LaTe₃.

In particular, the delay of the simulated fast recovery mentioned by the referee points towards an even longer-persisting enhancement of the critical temperature for intermediate fluences – which further stresses our underlying finding of a transiently enhanced T_c^* . An improved fit can be achieved by an increase of the thermalization time τ_{ph-ph} by few 10 % – a conceivable (fluence-dependent) deviation from the value determined for LaTe₃. However, as our model captures all experimentally observed trends, we believe that keeping the number of free parameters at a minimum provides a better understanding than a best fit.

tdGL simulations using the equilibrium critical temperature T_c . (a) Experimental data analog to Fig. 2f and (b) to Fig. 3a of the main manuscript. The tdGL simulations are performed using the parameters as described above, however, using a constant critical temperature of $T_c = 336$ K.

I would like to reiterate that the time-resolved ARPES data appears of the highest quality, is well presented, and the experimental observation of the increased transition temperature out-of-equilibrium is novel and exciting. In my opinion, the lack of microscopic insights into the model remains underwhelming.

We present a novel microscopic mechanism that explains the observed non-equilibrium behavior, see our response to the referee’s first comment. Further, we substantiate the proposed mechanism using a time-dependent variation of Ginzburg-Landau theory, which uses the electronic temperatures as an input parameter and captures all observed trends over a large fluence- and delay-range. We believe this significantly advances the understanding of collective excitations of such broken-symmetry ground states, as we are able to map in detail the underlying transient potential energy landscape and connect our observations to the energy-flow in the system. We believe that combining this with time-dependent structural information allows us to draw a comprehensive picture.

Referee report 3:

The authors have extensively replied to all questions and made occasional adaptations in the manuscript. I find the scarce changes still acceptable and I support publication of the work as is.

We thank the referee for his/her positive comments.

Further changes:

- Corrected axis labels in Supp. Fig. S8 (previously Fig. S7).
- Replaced a preprint reference (Trigo et al., arXiv:2006.08879 (2020)) by the published version (Trigo et al., Physical Review B 103(5), 054109 (2021)).

References

- [1] Lutz Waldecker, Roman Bertoni, Ralph Ernstorfer, and Jan Vorberger. Electron-phonon coupling and energy flow in a simple metal beyond the two-temperature approximation. Physical Review X, 6(2):021003, 2016.